# ⚛ LoraHub: Efficient Cross-Task Generalization via Dynamic LoRA Composition

**Chengsong Huang[‡§]\*, Qian Liu[†]\*, Bill Yuchen Lin[◊]\*, Tianyu Pang[†], Chao Du[†], Min Lin[†]**

[†]Sea AI Lab, Singapore
[§]Washington University in St. Louis, MO, USA
[◊]Allen Institute for AI, Seattle, WA, USA

## Abstract

Low-rank adaptations (LoRA) are often employed to fine-tune large language models (LLMs) for new tasks. This paper investigates LoRA composability for cross-task generalization and introduces LoraHub, a simple framework devised for the purposive assembly of LoRA modules trained on diverse given tasks, with the objective of achieving adaptable performance on unseen tasks. With just a few examples from a new task, LoraHub can fluidly combine multiple LoRA modules, eliminating the need for human expertise and assumptions. Notably, the composition requires neither additional model parameters nor gradients. Empirical results on the Big-Bench Hard benchmark suggest that LoraHub, while not surpassing the performance of in-context learning, offers a notable performance-efficiency trade-off in few-shot scenarios by employing a significantly reduced number of tokens per example during inference. Notably, LoraHub establishes a better upper bound compared to in-context learning when paired with different demonstration examples, demonstrating its potential for future development. Our vision is to establish a platform for LoRA modules, empowering users to share their trained LoRA modules. This collaborative approach facilitates the seamless application of LoRA modules to novel tasks, contributing to an adaptive ecosystem. Our code is available at `github.com/sail-sg/lorahub`, and all the pre-trained LoRA modules are released at `huggingface.co/lorahub`.

## 1 Introduction

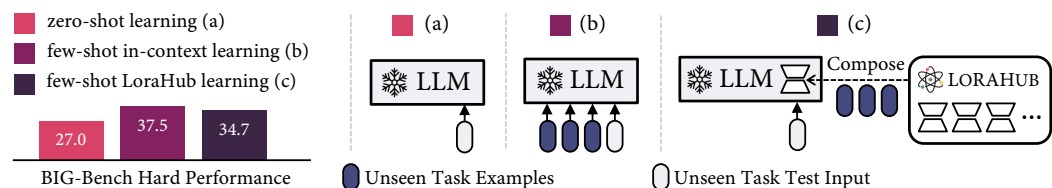

Figure 1: The illustration of zero-shot learning, few-shot in-context learning and few-shot LoraHub learning (ours). Note that the Compose procedure is conducted per task rather than per example. Our method achieves similar inference throughput as zero-shot learning, yet approaches the performance of in-context learning on the BIG-Bench Hard (BBH) benchmark.

Recent progress in natural language processing (NLP) has been largely fueled by large language models (LLMs) such as OpenAI GPT (Brown et al., 2020), FLAN-T5 (Chung et al., 2022), and LLaMA (Touvron et al., 2023). These models demonstrate top-tier performance

---

\*The first three authors contributed equally to this work. Correspondence to Qian Liu at `liuqian@sea.com`.

across different NLP tasks. However, their enormous parameter size presents issues regarding computational efficiency and memory usage during fine-tuning. To mitigate these challenges, Low-Rank Adaptation (LoRA) (Hu et al., 2022) has emerged as a parameter-efficient fine-tuning technique (Lester et al., 2021; He et al., 2022; An et al., 2022). By reducing memory demands and computational costs, it speeds up LLM training. LoRA achieves this by freezing the base model parameters (that is, an LLM) and training a lightweight module, which regularly delivers high performance on target tasks.

While prior research has targeted the efficiency enhancement facilitated by LoRA, there is a dearth of investigation into the inherent modularity and composability of LoRA modules. Typically, previous methods train LoRA modules to specialize in individual tasks. Yet, the intrinsic modularity of LoRA modules presents an intriguing research question: ***Would it be possible to compose LoRA modules to generalize to novel tasks in an efficient manner?*** In this paper, we tap into the potential of LoRA modularity for broad task generalization, going beyond single-task training to meticulously compose LoRA modules for malleable performance on unknown tasks. Crucially, our method enables an automatic assembling of LoRA modules, eliminating dependency on manual design or human expertise. With just a handful of examples from new tasks (e.g., 5), our approach can autonomously compose compatible LoRA modules without human intrusion. We do not make assumptions about which LoRA modules trained on particular tasks can be combined, allowing for flexibility in amalgamating any modules as long as they conform to the specification (e.g., using the same LLM). As our approach leverages several available LoRA modules, we refer to it as LoraHub and denote our learning method as **LoraHub learning**.

To validate the efficiency of our proposed methods, we test our approaches using the widely recognized BBH benchmark with FLAN-T5 (Chung et al., 2022) serving as the base LLM. The results underline the effectiveness of the LoRA module composition for unfamiliar tasks through a few-shot LoraHub learning process. Notably, our methodology achieves an average performance that closely matches that of few-shot in-context learning, while demonstrating a superior upper bound, particularly when using different demonstration examples. Additionally, our method substantially reduces the inference cost compared to in-context learning, eliminating the requirement of examples as inputs for the LLM. With fewer tokens per example during inference, our method significantly reduces computational overhead and enables faster responses. It aligns with a broader research trend, where recent studies are actively exploring approaches to reduce the number of input tokens (Zhou et al., 2023; Ge et al., 2023; Chevalier et al., 2023; Jiang et al., 2023a; Li et al., 2023; Jiang et al., 2023b). Our learning procedure is also notable for its computational efficiency, using a *gradient-free* approach to obtain the coefficients of LoRA modules and requiring only a handful of inference steps for unseen tasks. For example, when applied to a new task in BBH, our methodology can deliver superior performance in less than a minute using a single A100 card.

Importantly, LoraHub learning can feasibly be accomplished with a CPU-only machine, requiring proficiency solely for processing LLM inference. In our pursuit to democratize artificial intelligence, we are taking an important step forward by envisioning the establishment of the LoRA platform. The platform would serve as a marketplace where users can seamlessly share and access well-trained LoRA modules for diverse applications. LoRA providers have the flexibility to freely share or sell their modules on the platform without compromising data privacy. Users, equipped with CPU capability, can leverage trained LoRA modules contributed by others through automated distribution and composition algorithms. This platform not only cultivates a repository of reusable LoRA modules with a myriad of capabilities but also sets the stage for cooperative AI development. It empowers the community to collectively enrich the LLM's capabilities through dynamic LoRA composition.

## 2 Problem Statement

**Large Language Models** We assume that a large language model $M_\theta$ is based on Transformer architecture (Vaswani et al., 2017) and has been pre-trained on a large-scale text cor-

pus. The model architecture can be either encoder-decoder (Raffel et al., 2020) or decoder-only (Brown et al., 2020). Also, $M_\theta$ could also have been fine-tuned with a large set of instruction-following datasets such as Flan Colleciton (Longpre et al., 2023) and Prompt-Source (Bach et al., 2022).

**Cross-Task Generalization** In real-world situations, users often desire an LLM to perform novel tasks that it has not encountered before — an ability widely known as *cross-task generalization*. Generally, cross-task generalization falls into two categories: zero-shot learning (Mishra et al., 2022; Sanh et al., 2022; Chung et al., 2022; OpenAI, 2022; Lin et al., 2022), which necessitates no labeled examples of the new task, and few-shot learning (Ye et al., 2021; Min et al., 2022) which demands a handful of labeled examples. Assume we have $N$ distinct *upstream tasks* that the LLM has been trained on, denoted as $\mathbb{T} = \{\mathcal{T}_1, ..., \mathcal{T}_N\}$. Our paper primarily focuses on the latter category, where for an unseen target task $\mathcal{T}' \notin \mathbb{T}$, users can only provide a limited set of labeled examples, $Q$. Our aim is to modify the model $M_\theta$ to adapt it to task $\mathcal{T}'$ using only $Q$. An intuitive method would be to fine-tune the weights of $M_\theta$ based on $Q$, yielding an updated model $M_\phi$ with enhanced performance on $\mathcal{T}'$. However, this approach is inefficient, time-consuming, and unstable when $Q$ is small.

**LoRA Tuning** LoRA is a parameter-efficient fine-tuning method (Hu et al., 2022), facilitates the adaptation of LLMs using lightweight modules, eliminating the need for fine-tuning the entire weights. LoRA tuning involves keeping the original model weights frozen while introducing trainable low-rank decomposition matrices as adapter modules into each layer of the model. Compared to the base LLM, this module possesses significantly fewer trainable parameters, paving the way for rapid adaptation using minimal examples. As such, LoRA tuning presents a resource-efficient technique to quickly adapt LLMs for new tasks with restricted training data. However, traditional LoRA methods primarily concentrate on training and testing within the same tasks (Gema et al., 2023), rather than venturing into few-shot cross-task generalization.

## 3 Methodology

In this section, we provide an overview of our proposed method. We then explain the LoRA tuning procedure in detail. Last, we introduce the procedure of our LoraHub learning, which consists of the COMPOSE stage and the ADAPT stage.

### 3.1 Method Overview

As depicted in Figure 2, we initially train LoRA modules on a variety of upstream tasks. Specifically, for $N$ distinct upstream tasks, we separately train $N$ LoRA modules, each represented as $m_i$ for task $\mathcal{T}_i \in \mathbb{T}$. Subsequently, for a new task $\mathcal{T}' \notin \mathbb{T}$, such as Boolean Expressions represented in Figure 2, its examples $Q$ are utilized to steer the LoraHub learning process. The LoraHub learning encapsulates two main phases: the COMPOSE phase and the ADAPT phase. In the COMPOSE phase, all available LoRA modules are combined into a single integrated module $\hat{m}$, using $\{w_1, w_2, \ldots, w_N\}$ as coefficients. Each $w_i$ is a scalar value that can take on positive or negative values, and the combination can be done in different ways. During the ADAPT phase, the combined LoRA module $\hat{m}$ is amalgamated with the LLM $M_\theta$, and its performance on few-shot examples from the new task $\mathcal{T}'$ is assessed. A gradient-free algorithm is subsequently deployed to update $w$, enhancing $\hat{m}$'s performance (e.g., loss) on the few-shot examples $Q$. Finally, after iterating through $K$ steps, the optimum performing LoRA module is applied to the LLM $M_\theta$, yielding the final LLM $M_\phi = \text{LoRA}(M_\theta, \hat{m})$. This serves as an effectively adjusted model for the unseen task $\mathcal{T}'$, which will then be deployed and not updated anymore.

### 3.2 LoRA tuning on upstream tasks

LoRA effectively minimizes the number of trainable parameters through the process of decomposing the attention weight matrix update of the LLM, denoted as $W_0 \in R^{d \times k}$, into

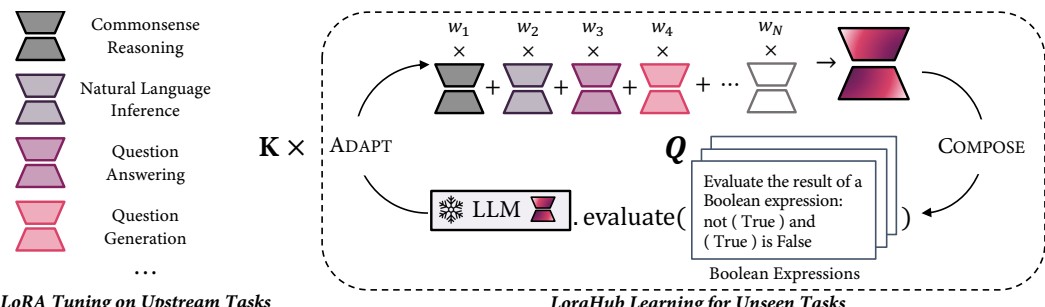

Figure 2: Our method encompasses two stages: the COMPOSE stage and the ADAPT stage. During the COMPOSE stage, existing LoRA modules are integrated into one unified module, employing a set of coefficients, denoted as $w$. In the ADAPT stage, the combined LoRA module is evaluated on a few examples from the unseen task. Subsequently, a gradient-free algorithm is applied to refine $w$. After executing $K$ iterations, a highly adapted combined LoRA module is produced, which can be incorporated with the LLM to perform the intended task.

low-rank matrices. In more specific terms, LoRA exhibits the updated weight matrix in the form $W_0 + \delta W = W_0 + AB$, where $A \in \mathbb{R}^{d \times r}$ and $B \in \mathbb{R}^{r \times k}$ are trainable low-rank matrices with rank $r$, a dimension significantly smaller than those of $d$ and $k$. In this context, the product $AB$ defines the LoRA module $m$, as previously elaborated. By leveraging the low-rank decomposition, LoRA substantially reduces the number of trainable parameters needed to adapt the weights of LLMs duriing fine-tuning.

### 3.3 COMPOSE: Element-wise composition of LoRA modules

Within the COMPOSE stage, we implement an element-wise method to combine LoRA modules. This process integrates the corresponding parameters of the LoRA modules, requiring the modules being combined to have the same rank $r$ to properly align the structures. Given that $m_i = A_i B_i$, the combined LoRA module $\hat{m}$ can be obtained by:

$$\hat{m} = (w_1 A_1 + w_2 A_2 + \cdots + w_N A_N)(w_1 B_1 + w_2 B_2 + \cdots + w_N B_N). \tag{1}$$

Notbly, as we show in Sec. 5, combining too many LoRA modules at once can expand the search space exponentially, which may destabilize the LoraHub learning process and prevent optimal performance. To mitigate this, we employ random selection to prune the candidate space, and more advanced pre-filtering algorithms could be explored in the future.

### 3.4 ADAPT: Weight optimization via gradient-free methods

During the ADAPT stage, our goal is to modify the coefficients $w$ to boost the model's performace on the examples from an unseen task. One might think of using gradient descent to optimize $w$, following standard backpropagation methods. However, this approach demands constructing a hypernetwork for all LoRA modules, similar to differentiable architecture search methods (Zhang et al., 2019). Constructing these hypernetworks demands for substantial GPU memory and time, posing a challenge. Given that $w$ consists of a relatively small number of parameters, we opted for gradient-free methods for optimization instead of gradient descent.

Inspired by previous work (Sun et al., 2022), we utilize a black-box optimization technique to find the optimal $w$. The optimization process is steered by the cross-entropy loss, setting the goal to locate the best set $\{w_1, w_2, \ldots, w_N\}$ that reduces the loss $L$ on the few-shot examples $Q$. Furthermore, we incorporate L1 regularization to penalize the sum of the absolute values of $w$, helping to prevent obtaining extreme values. Consequently, the final objective of LoraHub is to minimize $L + \alpha \cdot \sum_{i=1}^{N} |w_i|$, where $\alpha$ serves as a hyperparameter.

In terms of the gradient-free method, we leverage Shiwa, a combinatorial optimization approach (Liu et al., 2020). Shiwa offers a variety of algorithms and chooses the most suitable optimization algorithm for different circumstances. In most of the forthcoming experimental setups, we primarily employ the Covariance Matrix Adaptive Evolution Strategies (CMA-ES) (Hansen & Ostermeier, 1996). CMA-ES, as a stochastic and population-based optimization algorithm, offers versatility in addressing a broad spectrum of optimization challenges. It dynamically adjusts a search distribution, which is defined by a covariance matrix. During each iteration, CMA-ES systematically updates both the mean and covariance of this distribution to optimize the target function. In our application, we employ this algorithm to mold the search space for $w$. Ultimately, we use it to identify the optimal $w$ by evaluating their performance on the few-shot examples from an unseen task.

## 4 Experimental Results

In this section, we provide details on our main experiments. First, we give an overview of the experimental setup and implementation details. Next, we present our findings along with the results.

### 4.1 Experimental setup

**Large Language Model**   In our main experiments, we employ FLAN-T5 (Chung et al., 2022), particularly FLAN-T5-large, as the base LLM. The model has shown impressive abilities to perform zero-shot and few-shot learning.

**Candidate LoRA Modules**   Our methodology requires a compendium of LoRA modules trained on preceding tasks. For parity with FLAN, we adopt the tasks utilized to instruct FLAN-T5, thereby incorporating nearly 200 distinct tasks and their corresponding instructions. Following this, we trained several LoRA modules as potential candidates. During each experimental sequence, we randomly select 20 LoRA modules from them as the candidate for our LoraHub learning.

**Dataset and evaluation**   Our method is evaluated using the Big-Bench Hard (BBH) benchmark, a well-established standard that consists of multiple-choice questions from a variety of domains. The benchmark consists of 27 different tasks, which are regarded to be challenging for language models. For all tasks, we employ the exact match (EM) as our evaluation metric.

**Baseline Setup**   To enhance the demonstration of our method's performance, we expanded our comparisons beyond the zero-shot and in-context learning settings. We specifically chose three representative gradient-based methods for comparison: full fine-tuning (FFT), LoRA tuning (LoRA) (Hu et al., 2022), and IA3 fine-tuning (IA3) (Liu et al., 2022). For all gradient-based methods, for a fair comparsion, we train for 40 epochs on the same three runs of 5 examples employed in our methods. In the case of FFT, a learning rate of 3e-5 is employed, whereas for IA3 and LoRA, we adopt a learning rate of 2e-4. We report the performance of each method on the test set at the end of training (averaged over three runs) without any model selection to avoid potential selection bias.

### 4.2 Main results

As shown in Table 1, our experimental results demonstarte the superior efficacy of our method in comparison to zero-shot learning while closely resembling the performance of in-context learning (ICL) in few-shot scenarios. This observation is derived from an average performance of three runs, each leveraging different few-shot examples. Importantly, our model utilizes an equivalent number of tokens as the zero-shot method, notably fewer than the count used by ICL. Although occasional performance fluctuations, our method consistently outperforms zero-shot learning in most tasks. In the era of LLMs, the input

Table 1: Experimental results of zero-shot learning (Zero), few-shot in-context learning (ICL), IA3 fine-tuning (IA3), LoRA tuning (LoRA), full fine-tuning (FFT) and our proposed few-shot LoraHub learning (LoraHub) on the BBH benchmark with FLAN-T5-large as the base LLM. We denote algorithmic tasks with the superscript § following previous work (Wu et al., 2023b). Note that we employ three runs, each leveraging different 5-shot examples per task, as demonstrations for all few-shot methods. The average performance of all methods is reported below, and the best performance of each few-shot method can be found in the Appendix B.

| Task | Zero | ICL$_{avg}$ | IA3$_{avg}$ | LoRA$_{avg}$ | FFT$_{avg}$ | LoraHub$_{avg}$ |
|---|---|---|---|---|---|---|
| Boolean Expressions | 54.0 | 59.6 | 56.2 | 56.0 | 62.2 | 55.5 |
| Causal Judgement | 57.5 | 59.4 | 60.2 | 55.6 | 57.5 | 54.3 |
| Date Understanding | 15.3 | 20.4 | 20.0 | 35.8 | 59.3 | 32.9 |
| Disambiguation | 0.0 | 69.1 | 0.0 | 68.0 | 68.2 | 45.2 |
| Dyck Languages | 1.3 | 0.9 | 4.2 | 22.2 | 19.5 | 1.0 |
| Formal Fallacies | 51.3 | 55.3 | 51.5 | 53.6 | 54.0 | 52.8 |
| Geometric Shapes | 6.7 | 19.6 | 14.7 | 24 | 31.1 | 7.4 |
| Hyperbaton | 6.7 | 71.8 | 49.3 | 55.3 | 77.3 | 62.8 |
| Logical Deduction§ (five objects) | 21.3 | 39.1 | 32.7 | 40.0 | 42.2 | 36.1 |
| Logical Deduction§ (seven objects) | 12.7 | 40.7 | 33.8 | 37.3 | 44.9 | 36.8 |
| Logical Deduction§ (three objects) | 0.0 | 51.6 | 8.5 | 53.6 | 52.9 | 45.7 |
| Movie Recommendation | 62.7 | 55.8 | 61.8 | 51.5 | 66.0 | 55.3 |
| Multistep Arithmetic | 0.7 | 0.7 | 0.7 | 0.2 | 0.0 | 0.4 |
| Navigate | 47.3 | 45.3 | 46.2 | 48.0 | 48.0 | 47.1 |
| Object Counting | 34.7 | 32.4 | 35.1 | 38.7 | 35.6 | 33.7 |
| Penguins in a Table | 43.5 | 41.3 | 45.0 | 36.2 | 31.9 | 35.9 |
| Reasoning about Colored Objects | 32.0 | 40.2 | 40.7 | 39.6 | 37.6 | 40.0 |
| Ruin Names | 23.3 | 19.3 | 24.4 | 37.8 | 61.3 | 24.4 |
| Salient Translation Error Detection | 37.3 | 47.3 | 37.1 | 16.0 | 16.2 | 36.0 |
| Snarks | 50.0 | 54.2 | 53.9 | 55.6 | 66.7 | 56.9 |
| Sports Understanding | 56.0 | 54.7 | 55.1 | 56.5 | 54.0 | 56.7 |
| Temporal Sequences | 16.7 | 25.1 | 18.2 | 25.1 | 37.8 | 18.2 |
| Tracking Shuffled Objects§ (five objects) | 12.0 | 12.0 | 12.0 | 13.8 | 16.9 | 12.3 |
| Tracking Shuffled Objects§ (seven objects) | 6.7 | 6.7 | 6.7 | 10.0 | 9.8 | 7.7 |
| Tracking Shuffled Objects§ (three objects) | 24.7 | 31.1 | 30.7 | 30.9 | 32.0 | 29.2 |
| Web of Lies | 54.0 | 53.8 | 54.2 | 52.7 | 48.2 | 50.1 |
| Word Sorting | 1.3 | 0.5 | 1.3 | 4.9 | 4.9 | 1.1 |
| Avg Performance Per Task | 27.0 | 37.3 | 31.6 | 37.7 | 42.1 | 34.7 |
| Avg Tokens Per Example | 111.6 | 597.8 | 111.6 | 111.6 | 111.6 | 111.6 |
| Gradient-based Training | No | No | Yes | Yes | Yes | No |

length is directly proportional to the inference cost, and thus LoraHub's ability to economize on input tokens while approaching the peak performance grows increasingly significant. Moreover, as shown in Appendix Table 4, the upper bound performance of our method across these runs can surpass ICL on 18 tasks, demonstrating its potential for future development.

Even when compared to certain gradient-based optimization methods, our approach consistently demonstrates competitive performance. For example, as depicted in Table 1, our method exhibits a notable improvement of 3.1% on average in contrast to the promising IA3 method. Nevertheless, we acknowledge that our approach still falls behind LoRA tuning and full fine-tuning, especially in tasks that exhibit significant deviation from the upstream task. Taking Dyck Languages as an example, both LoraHub and ICL achieve

only an average performance of nearly 1.0% on these tasks, while LoRA and FFT methods showcase impressive results with only 5 examples.

### 4.3 Discussion

LoraHub addresses the challenge of reducing inference costs by eliminating the need for processing additional tokens, resulting in a noticeable reduction in overall inference expenses. However, it introduces an inherent cost during the ADAPT stage, necessitating extra inference steps, such as the 40 steps employed in our experiments. This introduces a trade-off between choosing the ICL approach and LoraHub, with the decision typically hinging on the nature of the situation.

For one-time ad-hoc tasks, the ICL approach should be more pragmatic due to LoraHub's additional inference step costs. In such scenarios, where immediate, single-use solutions are preferred, the simplicity and efficiency of ICL might outweigh the benefits of potential savings offered by LoraHub. Conversely, for recurring or similar tasks, LoraHub emerges as a compelling option. Despite the added inference step cost, LoraHub's ability to efficiently handle repetitive tasks, often occurring thousands of times, while concurrently reducing overall expenses, positions it as a viable option in such kind of situations.

In summary, our intention is not to replace ICL, but to present LoraHub as a complementary strategy with performance-efficiency trade-offs. Thus, we encourage a careful consideration of specific use cases and requirements when choosing between ICL and LoraHub, recognizing that the optimal solution may vary based on the nature and frequency of the tasks at hand.

## 5 Experimental Analysis

In this section, we thoroughly examine the characteristics of our proposed method and uncover several insightful findings. If not specified, we use FLAN-T5-large for all analysis.

> Does composing LoRA modules extend beyond the single module's benefits?

We acknowledge the investigation of cross-task performance in prior work (Jang et al., 2023), which delved into the capabilities of LoRA and proposed a novel method centered around LoRA module retrieval. In order to ensure a fair comparison, we conducted an experiment where we designed a LoRA retrieval mechanism based on the loss derived from few-shot examples.

Table 2: The average performance of various methods across all tasks in the benchmark BBH.

| LoRA Retrieval | LoraHub $_{avg}$ | LoraHub $_{best}$ |
|---|---|---|
| 31.7 | 34.7 | 41.2 |

Specifically, we ranked all LoRA module candidates according to this loss and evaluated the best candidate on the test set of the unseen task. As depicted in Table 2, the performance of LoRA retrieval is notably impressive, positioning it as a strong baseline. However, in comparison to LoraHub, the performance of LoRA retrieval is relatively less favorable

> How effective is the gradient-free optimization method?

To assess the effectiveness of our gradient-free optimization method in correctly identifying the most suitable LoRA module for a given downstream task, we carried out an empirical study using the WikiTableQuestions (Pasupat & Liang, 2015) (WTQ) dataset. We strategically included a LoRA module that was specifically trained on the WTQ dataset into our pool of LoRA candidate modules, which originally stemmed from tasks exclusive to the Flan Collection. Subsequently, we designated WTQ as the targeted downstream task and computed the weights consistent with the methods employed in LoraHub learning. As an end result, the WTQ-specific LoRA module was awarded the highest weight, ex-

emplifying the algorithm's success in recognizing it as the most relevant. Moreover, the combined LoRA module demonstrated marginal superiority over the WTQ LoRA module. This underscores the claim that the gradient-free optimization method has the ability to proficiently select the optimal upstream LoRA module for an unseen task.

> **Can LoraHub work well on non-instruction-tuning models?**

In previous investigations, we primarily focused on models with zero-shot capabilities that were trained with instruction tuning. However, for models like T5 without zero-shot abilities, where training has a larger effect on parameters, it was unclear if LoraHub could still effectively manage and improve them. Our experiments show that although these models perform worse than FLAN-T5, LoraHub learning can still enable them to effectively generrlize to unseen tasks. See Appendix C for more details.

> **Will the rank of LoRA modules impact the performance of LoraHub learning?**

The parameter rank plays a crucial role in the LoRA framework, directly influencing the number of trainable parameters utilized during LoRA tuning. This prompts an intriguing question: does the variation in rank values influence the outcomes observed within the LoraHub learning? Our analysis indicates that, for FLAN-T5, the choice of rank has minimal impact. However, for T5, it still exerts some influence. Empirical findings reveal that, in comparison to rank values of 4 or 64, a rank value of 16 consistently demonstrates superior performance across different runs, both in terms of average and optimal values. Additional results are available in Appendix C.

> **Does more LoRA modules lead to better results?**

In our main experiments, we randomly selected 20 LoRA modules for LoraHub learning. Therefore, we conducted experiments to investigate the effect of using different numbers of LoRA modules. The results demonstrate that as we increased the number of LoRA modules, the variance in performance increased. However, the maximum achievable performance also improved. More analysis on the variance and the detailed results can be found in Appendix H.

> **How much computational resource can be saved?**

We follow to the memory test settings from the LoRA-FA (Zhang et al., 2023b) study for an accurate benchmark. In this context, full fine-tuning required about 40GB of memory, whereas LoRA fine-tuning used around 34GB. Remarkably, LoraHub only utilized about 5GB of memory, illustrating its efficiency due to the inference-only mode, which eliminates the need for storing gradients and optimization states.

## 6 Related work

**Model Merging**   Our method substantially draws on the concept of LoRA module composition, and thus, aligns with the significant thread of research in model merging. This research focus is broadly categorized based on the ultimate objectives of model merging.

The first category focuses on merging entire models, and the goal is to combine individually trained models to approximate the performance benefits of model ensembling or multi-task learning. Prior works (Matena & Raffel, 2021; Jin et al., 2023; Yadav et al., 2023; Wu et al., 2023a) operated under the assumption of shared model architectures. For example, Matena & Raffel (2021) amalgamates models by approximating Gaussian posterior distributions garnered from Fisher information, while Yadav et al. (2023) merges models via resolving model interferences. Another approach is merging models with different architectures. For instance, Ainsworth et al. (2023) configures weights of different models prior to their merger. Following this objective, Stoica et al. (2023) merges models operating

on varying tasks by identifying common features, without requiring additional training. Unlike these works, our work focuses on merging models for better cross-task generalization.

The second category most closely aligns with our research, stemming from a shared motivation of module composition. Various scholars have made advances in this line of research: Kingetsu et al. (2021) decomposes and recomposes modules on the basis of their functionality; Ilharco et al. (2023) proposes modulating model behavior using task vectors; Lv et al. (2023) amalgamates parameter-efficient modules weighted according to task similarity; Zhang et al. (2023a) crafts modules by employing specific arithmetic operations; Sun et al. (2023) improves few-shot performance of unseen tasks by multi-task pre-training of prompts; Chronopoulou et al. (2023) averages adapter weights intended for transfer; Ponti et al. (2023) focuses on jointly learning adapters and a routing function that allocates skills to each task; and Muqeeth et al. (2023) concentrates on amalgamating experts in mixture of experts models; However, these methods generally necessitate multi-task training or human prior on module selection for the downstream task. In contrast, our method does not impose any special training requirements and simply employs vanilla LoRA tuning. Additionally, the module selection for downstream tasks is entirely data-driven without human prior knowledge. This design gives the advantage of easily adding new LoRA modules for reuse, allowing our method to flexibly scale up the number of LoRA module candidates in the future.

**Mixture of Experts**   The Mixture of Experts (MoE) is an ensemble method, often visualized as a collection of sub-modules, or "experts", each specializing in processing different types of input data. Each expert in this system is controlled by a unique gating network, activated based on the distinct nature of the input data. For every token in these input sequences, this network identifies and engages the most suitable experts to process the data. As a result, the performance is superior compared to relying on a single, generic model for all types of input. This technique has proven instrumental in numerous domains, such as natural language processing and computer vision (Jacobs et al., 1991; Shazeer et al., 2017; Du et al., 2022; Zhang et al., 2022; Wang et al., 2022; crumb, 2023). Our methodology displays similarities to MoE, wherein upstream-trained LoRA modules can be aligned with MoE's expert design. A noteworthy distinguishing factor is that our approach mechanism does not require any specialized manipulation of LoRAs during training while facilitating dynamic LoRA module assembly at any scale, each pre-tuned to different tasks. In contrast, MoE mandates a predetermined count of experts during both the training and testing phases. Recent studies on the interrelation between MoE and instruction tuning have demonstrated that the simultaneous application of both approaches enhances the effectiveness of each individually (Shen et al., 2023).

**Cross-Task generalization**   Recent advancements like CrossFit (Ye et al., 2021), ExT5 (Aribandi et al., 2022), FLAN (Wei et al., 2022), T0 (Sanh et al., 2022), Instruct-GPT (Ouyang et al., 2022), and ReCross (Lin et al., 2022) have been striving to foster a vastly multi-task model's generalization across different tasks, very much aligned with the objectives of our research. Among this cohort, the connections of CrossFit and ReCross with LoraHub are particularly noteworthy. The CrossFit framework (Ye et al., 2021) mandates a minimal number of labeled examples of the target task for few-shot fine-tuning. However, its limitation lies in the application of task names as hard prefixes in templates, posing challenges in the task's generalization. On the other hand, while ReCross mitigates the need for labels in few-shot examples for retrieval, it necessitates a fine-tuning process using the retrieved data. This procedure appears time-consuming when compared to LoraHub's approach. Through the deployment of few-shot labeled examples and a gradient-free optimization process, LoraHub facilitates an iterative update of weights to compose the LoRA modules. The resultant method is more efficient and cost-effective relative to previous work. Overall, LoraHub offers a more practical and viable solution to the optimization process.

# 7 Conclusion

In this work, we have introduced LoraHub, a strategic framework for composing LoRA modules trained on diverse tasks in order to achieve adaptable performance on new tasks. Our approach enables the fluid combination of multiple LoRA modules using just a few examples from a novel task, without requiring additional model parameters or human expertise. The empirical results on the BBH benchmark demonstrate that LoraHub can effectively match the performance of in-context learning in few-shot scenarios, removing the need for in-context examples during inference. Overall, our work shows the promise of strategic LoRA composability for rapidly adapting LLMs to diverse tasks. By fostering reuse and combination of LoRA modules, we can work towards more general and adaptable LLMs while minimizing training costs.

## Reproducibility Statement

The authors have made great efforts to ensure the reproducibility of the empirical results reported in this paper. Firstly, the experiment settings, evaluation metrics, and datasets were described in detail in Section 4.1. Secondly, the codes and script for reproduce the result will be opensource after accepted. Second, the source code implementing the proposed method and experiments will be made publicly available at upon acceptance of the paper. Third, pre-trained LoRA modules from this work along with their configuration files and weights will be shared. These allow reproduction without retraining the LoRA modules, enabling quick testing and verification.

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

Table 3: The top five beneficial LoRA modules for BBH tasks and their associated upstream tasks, the average weight values and the average performance on all BBH tasks.

| Rank | Dataset: Task | Weight | Perf | Task Description |
|------|---------------|--------|------|------------------|
| 1 | WIQA: Last Process | 0.72 | 28.1 | Identifying the last step of a given process. |
| 2 | RACE: Is this the Right Answer | 0.68 | 30.8 | Determining if given answer is correct. |
| 3 | WIQA: First Process | 0.63 | 28.1 | Identifying the first step of a given process. |
| 4 | AdversarialQA: BiDAF | 0.61 | 25.1 | Answering question created by an adversarial model-in-the-loop. |
| 5 | WebQuestions: What is the Answer | 0.58 | 27.0 | Answering question based on information extracted from the web. |

## A  More Analysis

**Which LoRA modules are most effective for BBH tasks?**

We hypothesized that the amalgamation of LoRA modules could incorporate skills and insights from a variety of specific tasks. To evaluate this, we examined the extent of influence a single LoRA module had amongst all tasks from the BBH benchmark. We measured the impact of each isolated task by calculating the average absolute weight. The top five modules, presented in Table 3, were found to have substantial influence, as indicated by their maximum average weights, which suggested that they were notably more effective in cross-task transfer. Remarkably, a common feature among these top five modules was their association with tasks requiring reading comprehension and reasoning skills—attributes indicative of higher cognitive complexity. However, it is worth noting that none of the modules exhibited consistent improvement across all BBH tasks, as reflected in their average performance on all BBH tasks, which did not show a significant improvement compared to the original FLAN-T5-large, except for the Rank 2. The results underscore the advantages of composing diverse modules in LoraHub.

**How effective is the gradient-free optimization method?**

To assess the effectiveness of our gradient-free optimization method in correctly identifying the most suitable LoRA module for a given downstream task, we carried out an empirical study using the WikiTableQuestions (Pasupat & Liang, 2015) (WTQ) dataset. We strategically included a LoRA module that was specifically trained on the WTQ dataset into our pool of LoRA candidate modules, which originally stemmed from tasks exclusive to the Flan Collection. Subsequently, we designated WTQ as the targeted downstream task and computed the weights consistent with the methods employed in LoraHub learning. As an end result, the WTQ-specific LoRA module was awarded the highest weight, exemplifying the algorithm's success in recognizing it as the most relevant. Moreover, the combined LoRA module demonstrated marginal superiority over the WTQ LoRA module. This underscores the claim that the gradient-free optimization method has the ability to proficiently select the optimal upstream LoRA module for an unseen task.

## B  Result of Best Results

As shown in Table 4, compared to gradient-based parameter-efficient training methods like LoRA and IA3, our approach demonstrates superior performance in terms of best results over experimental runs. While it exhibits a noticeable lag behind the fully fine-tuning (FFT) method, which updates all parameters during training, this observation suggests that our proposed method has a promising upper limit. We anticipate that future research efforts can contribute to accelerating the optimization speed and further enhancing the efficacy of our approach.

Table 4: Experimental results of several few-shot methods, including in-context learning (ICL), IA3 fine-tuning (IA3), LoRA tuning (LoRA), full fine-tuning (FFT) and our LoraHub learning (LoraHub) on the BBH benchmark with FLAN-T5-large as the base LLM. We denote algorithmic tasks with the superscript § following previous work (Wu et al., 2023b). Note that we use 5 examples per task as the demonstration for all methods. The best (*best*) performance is reported as the maximum value obtained across three runs.

| Task | $ICL_{best}$ | $IA3_{best}$ | $LoRA_{best}$ | $FFT_{best}$ | $LoraHub_{best}$ |
|---|---|---|---|---|---|
| Boolean Expressions | 62.7 | 58.0 | 60.7 | 65.3 | 60.7 |
| Causal Judgement | 59.8 | 62.1 | 57.5 | 60.9 | 63.2 |
| Date Understanding | 21.3 | 20.7 | 40.7 | 67.3 | 45.3 |
| Disambiguation | 69.3 | 0.0 | 68.7 | 70.7 | 68.0 |
| Dyck Languages | 2.0 | 4.7 | 25.3 | 33.3 | 2.7 |
| Formal Fallacies | 59.3 | 52.0 | 56.7 | 56.0 | 59.3 |
| Geometric Shapes | 20.0 | 15.3 | 28.7 | 39.3 | 18.7 |
| Hyperbaton | 72.7 | 49.3 | 57.3 | 82.0 | 72.7 |
| Logical Deduction§ (five objects) | 39.3 | 32.7 | 41.3 | 43.3 | 40.0 |
| Logical Deduction§ (seven objects) | 42.0 | 34.0 | 42.7 | 46.0 | 46.0 |
| Logical Deduction§ (three objects) | 52.7 | 8.7 | 56.7 | 60.7 | 52.7 |
| Movie Recommendation | 56.7 | 62.0 | 64.5 | 70.7 | 62.0 |
| Multistep Arithmetic | 0.7 | 0.7 | 0.7 | 0.0 | 1.3 |
| Navigate | 46.7 | 47.3 | 50.7 | 50.0 | 51.3 |
| Object Counting | 34.7 | 35.3 | 42.0 | 38.0 | 36.7 |
| Penguins in a Table | 43.5 | 45.7 | 41.3 | 37.0 | 47.8 |
| Reasoning about Colored Objects | 41.3 | 41.3 | 40.7 | 38.7 | 44.7 |
| Ruin Names | 20.7 | 25.3 | 42.0 | 66.0 | 28.7 |
| Salient Translation Error Detection | 48.0 | 37.3 | 17.3 | 21.3 | 42.7 |
| Snarks | 55.1 | 56.4 | 59.0 | 69.2 | 61.5 |
| Sports Understanding | 56.7 | 55.3 | 58.7 | 58.7 | 62.7 |
| Temporal Sequences | 26.7 | 18.7 | 31.3 | 48.7 | 21.3 |
| Tracking Shuffled Objects§ (five objects) | 12.0 | 12.0 | 16.0 | 20.0 | 16.7 |
| Tracking Shuffled Objects§ (seven objects) | 6.7 | 6.7 | 12.0 | 10.0 | 15.3 |
| Tracking Shuffled Objects§ (three objects) | 31.3 | 30.7 | 32.0 | 36.0 | 31.3 |
| Web of Lies | 54.0 | 54.7 | 55.3 | 54.0 | 57.3 |
| Word Sorting | 0.7 | 1.3 | 5.3 | 6.0 | 1.3 |
| Best Performance (Average) | 38.4 | 32.1 | 40.9 | 46.2 | 41.2 |

# C  Result of non-instrcution-tuned models

Table 5: Comparsion among different ranks for few-shot LoraHub learning with the backbone T5-large (Raffel et al., 2020) on the BBH benchmark. Note that the T5-large model achieved 0.0% on all tasks under the zero-shot setting except *Dyck Languages*, where it scored 0.67%.

| Task ↓     Rank → | $4_{avg}$ | $4_{best}$ | $16_{avg}$ | $16_{best}$ | $64_{avg}$ | $64_{best}$ |
|---|---|---|---|---|---|---|
| Boolean Expressions | 52.13 | 57.33 | 50.67 | **58.00** | 47.47 | 58.00 |
| Causal Judgement | 52.41 | **55.17** | 49.66 | 54.02 | 50.80 | 54.02 |
| Date Understanding | 0.40 | 2.00 | 14.40 | **29.33** | 4.53 | 10.00 |
| Disambiguation | 10.00 | 31.33 | 26.93 | **42.00** | 1.73 | 4.67 |
| Dyck Languages | 0.40 | 0.67 | 0.40 | 0.67 | 0.40 | **2.00** |
| Formal Fallacies | 48.40 | **54.00** | 46.93 | 51.33 | 46.93 | 50.00 |
| Geometric Shapes | 0.00 | 0.00 | 6.53 | **32.67** | 1.47 | 7.33 |
| Hyperbaton | 30.13 | 50.00 | 39.07 | **57.33** | 32.93 | 48.00 |
| Logical Deduction[§] (five objects) | 5.20 | 14.67 | 8.80 | **19.33** | 1.33 | 6.67 |
| Logical Deduction[§] (seven objects) | 6.40 | 17.33 | 9.33 | **19.33** | 3.47 | 16.00 |
| Logical Deduction[§] (three objects) | 14.40 | 32.00 | 21.73 | **34.67** | 6.93 | 15.33 |
| Movie Recommendation | 7.07 | 18.67 | 7.87 | **22.00** | 1.20 | 6.00 |
| Multistep Arithmetic two | 0.00 | 0.00 | 0.00 | 0.00 | 0.00 | 0.00 |
| Navigate | 49.60 | 54.67 | 52.27 | **56.67** | 49.87 | 52.00 |
| Object Counting | 7.20 | 18.00 | 16.00 | 21.33 | 13.73 | **26.67** |
| Penguins in a Table | 6.52 | 13.04 | 10.43 | **17.39** | 0.43 | 2.17 |
| Reasoning about Colored Objects | 6.27 | 10.00 | 5.07 | **16.67** | 0.53 | 2.67 |
| Ruin Names | 7.73 | 13.33 | 13.20 | **28.00** | 5.73 | 15.33 |
| Salient Translation Error Detection | 0.00 | 0.00 | 1.73 | **8.67** | 0.00 | 0.00 |
| Snarks | 21.28 | 42.31 | 49.49 | **60.26** | 16.15 | 38.46 |
| Sports Understanding | 46.53 | **58.67** | 46.80 | 58.67 | 46.53 | 58.67 |
| Temporal Sequences | 3.07 | 13.33 | 6.53 | **26.67** | 2.40 | 12.00 |
| Tracking Shuffled Objects[§] (five objects) | 5.20 | **14.00** | 4.13 | 9.33 | 0.13 | 0.67 |
| Tracking Shuffled Objects[§] (seven objects) | 2.67 | 10.00 | 2.80 | **14.00** | 3.20 | 8.00 |
| Tracking Shuffled Objects[§] (three objects) | 3.73 | 17.33 | 16.27 | **34.67** | 5.87 | 26.67 |
| Web of Lies | 48.53 | 54.00 | 54.00 | 56.00 | 54.67 | **57.33** |
| Word Sorting | 0.40 | **0.67** | 0.13 | 0.67 | 0.00 | 0.00 |
| Average Performance per Task | 16.14 | 24.17 | 20.78 | **30.73** | 14.76 | 21.43 |

# D    Result of larger model

Table 6: Experimental results of zero-shot learning (Zero) and our few-shot LoraHub learning (LoraHub) on the BBH benchmark with FLAN-T5-xl as the base LLM. Note that we use 5 examples per task as the demonstration for both ICL and LoraHub. The average (*avg*) performance of LoraHub is computed over 5 runs with different random seeds, while the best (*best*) performance is reported as the maximum value obtained across these runs. We can see the trend of the results are similar to FLAN-T5-large.

| Task | Zero | LoraHub $_{avg}$ | LoraHub $_{best}$ |
|---|---|---|---|
| Boolean Expressions | 52.0 | 58.7 | 63.3 |
| Causal Judgement | 62.1 | 53.8 | 59.8 |
| Date Understanding | 38.0 | 37.6 | 38.0 |
| Disambiguation Qa | 0.0 | 20.5 | 54.7 |
| Dyck Languages | 1.3 | 0.9 | 2.0 |
| Formal Fallacies | 56.0 | 56.0 | 56.0 |
| Geometric Shapes | 8.7 | 17.5 | 28.0 |
| Hyperbaton | 45.3 | 53.5 | 56.7 |
| Logical Deduction[§] (five objects) | 1.3 | 42.7 | 48.7 |
| Logical Deduction[§] (seven objects) | 8.7 | 44.3 | 50.0 |
| Logical Deduction[§] (three objects) | 0.7 | 56.4 | 61.3 |
| Movie Recommendation | 2.0 | 62.8 | 66.0 |
| Multistep Arithmetic Two | 0.0 | 0.4 | 0.7 |
| Navigate | 50.7 | 50.7 | 50.7 |
| Object Counting | 39.3 | 40.7 | 48.0 |
| Penguins In A Table | 17.4 | 40.9 | 45.7 |
| Reasoning About Colored Objects | 46.7 | 47.3 | 50.7 |
| Ruin Names | 18.0 | 35.6 | 44.7 |
| Salient Translation Error Detection | 44.7 | 45.1 | 48.7 |
| Snarks | 60.3 | 60.8 | 61.5 |
| Sports Understanding | 56.7 | 51.3 | 53.3 |
| Temporal Sequences | 21.3 | 21.5 | 22.0 |
| Tracking Shuffled Objects[§] (five objects) | 3.3 | 9.9 | 13.3 |
| Tracking Shuffled Objects[§] (seven objects) | 5.3 | 7.3 | 8.7 |
| Tracking Shuffled Objects[§] (three objects) | 7.3 | 21.7 | 31.3 |
| Web Of Lies | 54.7 | 47.1 | 48.7 |
| Word Sorting | 1.3 | 1.5 | 2.0 |
| Average Performance per Task | 25.8 | 36.5 | 41.3 |

# E   Improving the Robustness of LoraHub

In order to enhance the robustness of LoraHub, we explored a straightforward approach in the selection of LoRA module candidates. Specifically, we first identified 20 LoRA module candidates with the lowest loss on the few-shot examples. Our findings indicate a slight improvement in overall performance after applying the pre-filtering startegy. Since the primary instability in our approach arises from the selection of LoRA candidates. This method involves choosing a fixed set of LoRA candidates to ensure the stability of our approach.

Table 7: The experimental results of loss-based pre-filtering.

| Task | LoraHub$_{\text{avg}}$ | LoraHub$_{\text{filter}}$ |
|---|---|---|
| Boolean Expressions | 55.5 | 60.00 |
| Causal Judgement | 54.3 | 52.9 |
| Date Understanding | 32.9 | 33.3 |
| Disambiguation | 45.2 | 62.7 |
| Dyck Languages | 1.0 | 0.0 |
| Formal Fallacies | 52.8 | 54.0 |
| Geometric Shapes | 7.4 | 4.0 |
| Hyperbaton | 62.8 | 64.0 |
| Logical Deduction[§] (five objects) | 36.1 | 37.3 |
| Logical Deduction[§] (seven objects) | 36.8 | 22.0 |
| Logical Deduction[§] (three objects) | 45.7 | 56.0 |
| Movie Recommendation | 55.3 | 68.0 |
| Multistep Arithmetic | 0.4 | 0.7 |
| Navigate | 47.1 | 49.3 |
| Object Counting | 33.7 | 38.7 |
| Penguins in a Table | 35.9 | 37.0 |
| Reasoning about Colored Objects | 40.0 | 33.3 |
| Ruin Names | 24.4 | 22.0 |
| Salient Translation Error Detection | 36.0 | 24.0 |
| Snarks | 56.9 | 52.66 |
| Sports Understanding | 56.7 | 58.0 |
| Temporal Sequences | 18.2 | 27.3 |
| Tracking Shuffled Objects[§] (five objects) | 12.3 | 11.3 |
| Tracking Shuffled Objects[§] (seven objects) | 7.7 | 8.0 |
| Tracking Shuffled Objects[§] (three objects) | 29.2 | 32.7 |
| Web of Lies | 50.1 | 46.0 |
| Word Sorting | 1.1 | 1.3 |
| Avg Performance Per Task | 34.7 | 35.4 |

# F   Performance on General Important Task

In our research, we have identified specific LoRA modules that exhibit significant impact when integrated into merged LoRAs. Our focus lies in assessing the performance of the top five task-related LoRAs on the BBH benchmark. The results indicate that these top LoRAs perform similarly or even worse than zero-shot in most cases. Only one of them stands out as significantly better than zero-shot. However, it's worth noting that this performance is not as impressive as Lorahub. These findings support the idea that the merging process can improve overall performance.

Table 8: Detailed experimental results of top five LoRA modules shown in Table 3 on BBH tasks.

| Task | WIQA: Last | RACE: Right | WIQA: First | ADQA | WebQA |
|------|-----------|-------------|-------------|------|-------|
| Boolean Expressions | 52.67 | 58.00 | 52.67 | 54.67 | 53.33 |
| Causal Judgement | 55.17 | 63.22 | 55.17 | 57.47 | 57.47 |
| Date Understanding | 17.33 | 19.33 | 17.33 | 16.67 | 15.33 |
| Disambiguation | 0.00 | 0.00 | 0.00 | 0.00 | 0.00 |
| Dyck Languages | 0.67 | 0.67 | 0.67 | 1.33 | 1.33 |
| Formal Fallacies | 51.33 | 51.33 | 51.33 | 51.33 | 51.33 |
| Geometric Shapes | 8.00 | 13.33 | 8.00 | 6.67 | 7.33 |
| Hyperbaton | 16.67 | 44.00 | 16.67 | 1.33 | 6.00 |
| Logical Deduction§ (five objects) | 23.33 | 28.00 | 23.33 | 19.33 | 20.67 |
| Logical Deduction§ (seven objects) | 22.00 | 26.00 | 22.00 | 10.67 | 12.00 |
| Logical Deduction§ (three objects) | 0.67 | 9.33 | 0.67 | 0.00 | 0.00 |
| Movie Recommendation | 63.33 | 62.67 | 63.33 | 56.67 | 63.33 |
| Multistep Arithmetic | 0.67 | 0.67 | 0.67 | 0.67 | 0.67 |
| Navigate | 47.33 | 50.00 | 47.33 | 47.33 | 47.33 |
| Object Counting | 34.67 | 34.00 | 34.67 | 35.33 | 35.33 |
| Penguins in a Table | 45.65 | 41.30 | 45.65 | 39.13 | 43.48 |
| Reasoning about Colored Objects | 40.00 | 37.33 | 40.00 | 31.33 | 30.67 |
| Ruin Names | 22.00 | 21.33 | 22.00 | 17.33 | 22.67 |
| Salient Translation Error Detection | 36.67 | 34.67 | 36.67 | 32.67 | 37.33 |
| Snarks | 52.56 | 55.13 | 52.56 | 47.44 | 52.56 |
| Sports Understanding | 56.00 | 58.67 | 56.00 | 55.33 | 55.33 |
| Temporal Sequences | 16.67 | 17.33 | 16.67 | 12.67 | 17.33 |
| Tracking Shuffled Objects§ (five objects) | 12.00 | 12.00 | 12.00 | 10.67 | 12.00 |
| Tracking Shuffled Objects§ (seven objects) | 6.67 | 6.67 | 6.67 | 6.67 | 6.67 |
| Tracking Shuffled Objects§ (three objects) | 20.67 | 30.67 | 20.67 | 10.67 | 25.33 |
| Web of Lies | 54.67 | 54.00 | 54.67 | 54.00 | 54.00 |
| Word Sorting | 1.33 | 1.33 | 1.33 | 1.33 | 1.33 |
| Avg Performance per Task | 28.10 | 30.78 | 28.10 | 25.14 | 27.04 |
| Δ FLAN-T5-large | 1.10 | 3.78 | 1.10 | -1.86 | 0.04 |

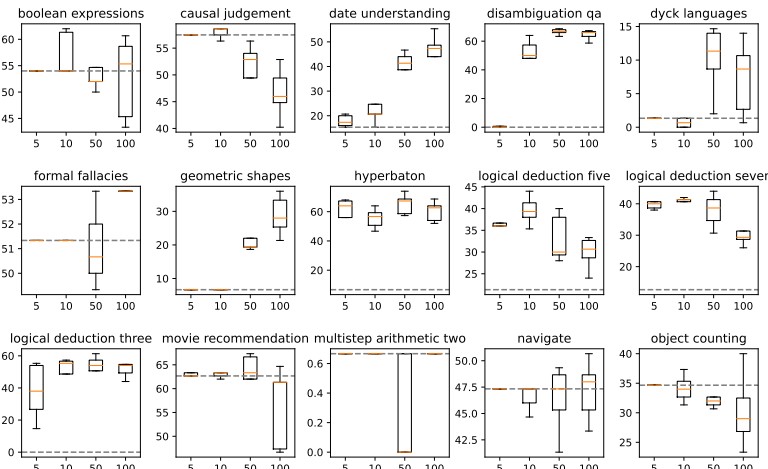

Figure 3: The influence of number of LoRA modules on 15 tasks from BBH, and each box is obtained from 5 separate runs. The horizontal axis shows the number of LoRA modules to be composed in LoraHub learning.

## G   Implementation details

We implemented LoRA tuning using the Huggingface PEFT library (Mangrulkar et al., 2022), with the rank being set as 16. The gradient-free method was implemented using the open-source Nevergrad optimization library (Rapin & Teytaud, 2018), with a constraint that the absolute value of LoRA weights should not exceed 1.5. Originally, all coefficients of LoRA modules were set at zero.

In our standard settings, we set the maximum number of iterations $K$ as 40. The same 5 examples were used during our LoraHub learning and the few-shot in-context learning. The hyperparameter $\alpha$ is set as 0.05. Regarding the hyperparameters for training candidate LoRA modules, we maintained consistency across all modules, setting the batch size at 64, the learning rate at $1e-4$, and the number of training epochs at 10.

## H   Influence of Number of LoRA modules

As shown in Figure 3, with an increase in the number of LoRA module candidates, there is a corresponding increase in the performance variance. Based on our in-depth analysis, the primary source of variance is not related to gradient-free optimization algorithms but rather associated with the LoRA candidate modules. In other words, once the candidates are determined, random seeds have minimal impact on the final performance. Hence, we posit that the observed instability primarily arises from the inherent challenge of balancing the quantity and quality of the LoRA module candidates.

## I   The Impact of Threshold

In this section, we omitted the threshold in our implementation, and the results are summarized in Table 9. Our observations indicate that the removal of the threshold had minimal impact on the majority of tasks, underscoring the robustness of the gradient-free optimization algorithm itself in most cases. The algorithm efficiently identified reasonable ranges even without specific upper and lower bounds. However, three tasks, namely Date Understanding, Disambiguation and Hyperbaton, exhibited notable effects. The resulting performance decline led to an average decrease of 1.2% compared to the setting with threshold.

This highlights the significance of establishing a reasonable threshold to mitigate extreme scenarios.

Table 9: The comparsion between LoraHub and LoraHub without threshold.

| Task | LoraHub$_{avg}$ with threshold | LoraHub$_{avg}$ without threshold |
|---|---|---|
| Boolean Expressions | 55.5 | 54.0 |
| Causal Judgement | 54.3 | 54.8 |
| Date Understanding | 32.9 | 17.7 |
| Disambiguation | 45.2 | 40.6 |
| Dyck Languages | 1.0 | 1.1 |
| Formal Fallacies | 52.8 | 51.7 |
| Geometric Shapes | 7.4 | 6.7 |
| Hyperbaton | 62.8 | 55.5 |
| Logical Deduction[§] (five objects) | 36.1 | 36.5 |
| Logical Deduction[§] (seven objects) | 36.8 | 35.6 |
| Logical Deduction[§] (three objects) | 45.7 | 49.9 |
| Movie Recommendation | 55.3 | 59.3 |
| Multistep Arithmetic | 0.4 | 0.7 |
| Navigate | 47.1 | 47.6 |
| Object Counting | 33.7 | 34.7 |
| Penguins in a Table | 35.9 | 33.8 |
| Reasoning about Colored Objects | 40.0 | 37.9 |
| Ruin Names | 24.4 | 24.0 |
| Salient Translation Error Detection | 36.0 | 37.1 |
| Snarks | 56.9 | 51.6 |
| Sports Understanding | 56.7 | 55.9 |
| Temporal Sequences | 18.2 | 16.7 |
| Tracking Shuffled Objects[§] (five objects) | 12.3 | 12.3 |
| Tracking Shuffled Objects[§] (seven objects) | 7.7 | 8.5 |
| Tracking Shuffled Objects[§] (three objects) | 29.2 | 29.8 |
| Web of Lies | 50.1 | 50.3 |
| Word Sorting | 1.1 | 1.3 |
| Avg Performance Per Task | 34.7 | 33.5 |

