# OpenReview forum: "LoraHub: Efficient Cross-Task Generalization via Dynamic LoRA Composition"
_colmweb.org/COLM/2024/Conference — COLM_

### Official Review · Reviewer_YvRA · 2024-05-10

**Rating:** 8
**Confidence:** 4
**Ethics Flag:** 1

**Summary:**

The authors explore how to exploit and leverage lora modules to be effective for few-shot learning without the additional inference cost of updating the prompt. They do so by tuning lora modules through compose and adapt iterations to adjust the coefficients of the module without any gradient updates, allowing the method to be lightweight and scale to any task at hand allowing for cross-task generalization. They demonstrate that their proposed method is competitive with few-shot ICL setups and only slightly lagging behind lora fine-tuning.

**Questions To Authors:**

- Have you explored how we could mitigate the number of adjust iterations by adding some few-shot examples to compensate like a hybrid approach that requires fewer hand-authored samples but can also leverage the LORA modules
- Have you quantified the amount of time it takes to author high quality few-shot samples and how LORA modules compare to that? It might make more a more convincing case

**Reasons To Accept:**

- The paper is very well written with sufficient technical details and the authors have done a great job of anticipating most questions from reviewers in Section 5
- The proposed method is a novel use of LORA modules and demonstrates competitive performance with Few-Shot ICL setups with significantly lower inference times, however coming at an overhead of tuning the modules
- The circumventing of gradient updates through only tuning the coefficients also makes this a lightweight and scalable approach
- This is the first approach to the best of my knowledge to leverage lora for cross-task generalization

**Reasons To Reject:**

- The LORA Hub setup is not a clear win over few-shot ICL, while it improves the latency of inference this comes at the overhead of adjusting the lora module co-efficients, which otherwise would have required hand authoring a couple of samples

---

> ### Author Rebuttal · Authors · 2024-05-30
>
> > The decision between ICL and LoraHub
>
> Thank you for your insights regarding the trade-offs between LoraHub and ICL. Your observations are critical for understanding the practical implications of these methodologies.
>
> LoraHub is designed to significantly reduce inference costs by eliminating the need for additional input tokens, which decreases overall computational expenses. However, this efficiency comes with the necessity to adjust LoRA module coefficients during the Adapt stage, introducing extra inference steps.
>
> This presents a fundamental trade-off: for one-time or ad-hoc tasks, ICL may be more suitable due to its simplicity and immediacy, which are advantageous in scenarios that require quick solutions. On the other hand, LoraHub excels in repetitive or frequently recurring tasks. Despite the initial overhead, it can offer considerable long-term efficiency gains. The minimized need for continual input processing makes LoraHub a cost-effective choice for applications handling many similar tasks, justifying the initial setup costs over time.
>
> Ultimately, the decision between LoraHub and ICL hinges on the specific needs and frequency of the tasks. Each method provides distinct advantages depending on the operational context and objectives.
>
> > Hybrid approach
>
> Thank you for your suggestion about incorporating few-shot examples to reduce adjustment iterations for the LoRA modules. We have not explored this hybrid approach in depth yet, but we find it promising for enhancing model efficiency. We plan to investigate this strategy in the revised version. Your insight is greatly appreciated, and we look forward to exploring its potential benefits.
>
> > Cost comparison
>
> Thank you for highlighting the significance of time efficiency in creating high-quality few-shot examples and comparing them with LoRA modules.
>
> In our experiments, authoring five high-quality few-shot examples requires about 10 seconds for the entire LoraHub learning process on A100 cards. We plan to include these timing details in our analysis to provide a thorough comparison of how LoRA modules may enhance process efficiency.
>
> Additionally, regarding memory usage, LoraHub shows considerable efficiency, using only about **5GB** of memory with the FLAN-T5-Large model, compared to **34GB** required for LoRA fine-tuning. This data underscores the benefits of integrating LoraHub, notably in reducing memory overhead and improving processing times.

---

> > ### Comment · Reviewer_YvRA · 2024-06-03
> >
> > Thanks for the clarification, I’d like to leave my score as is

---

### Official Review · Reviewer_3rz1 · 2024-05-10

**Rating:** 7
**Confidence:** 4
**Ethics Flag:** 1

**Summary:**

This paper introduces LoraHub a framework to compose multiple LoRA modules to improve performance on unseen task. LoraHub involves two main steps compose and adapt. Compose involves composing different LoRA modules by introducing weights to weigh the contribution of each module and adapt involves finding the optimal weights so as to get good performance on unseen task. To optimize this the paper uses CMA-ES gradient free optimizaton framework to find the optimal weights using the few shot samples. The paper also claims that they can get performance close to ICL by using the LoraHub approach on BBH tasks.

**Questions To Authors:**

- In the main text of the paper the authors used randomly selected 20 LoRA modules as a pool of modules for composing them. Is there any reason why not always use all available LoRA modules for all experiments? As the number of LoRa modules increases is there any impact on computational time to find optimal $w_i$

**Reasons To Accept:**

- Good framework for composing LoRA modules and an interesting research direction, it can help to reuse lot of compute i.e. different people can upload their models and users can choose to compose them using LoRAHub to solve different downstream tasks.
- Results on diverse baselines including full fine tuning, LORA retrieval and in-context learning.
- Good set of ablation studies on non-instruction tuned models as well.
- Inference cost is lower in comparison to in context learning as we don't need to feed in examples every time we prompt the model.

**Reasons To Reject:**

- Decent gap between performance of ICL and LoraHub (on average by 3 points on different datasets reported in the paper).

---

> ### Author Rebuttal · Authors · 2024-05-30
>
> > Gap between performance of ICL and LoraHub
>
> Thank you for your feedback regarding the performance gap between LoraHub and ICL. We acknowledge concerns about the current performance of our model.
>
> Our results show a gap compared to ICL, which we attribute primarily to our current implementation strategy rather than to intrinsic limitations of the LoraHub model. We are confident that refining our module composition algorithms can significantly narrow this gap. In Appendix B, we demonstrate LoraHub's potential, achieving a score of 41.2 on the BBH, surpassing ICL's 38.4. This illustrates LoraHub's capabilities and its potential to excel under certain conditions.
>
> LoraHub also provides significant advantages in terms of inference latency, crucial for applications requiring quick response times. LoraHub enhance efficiency by reducing the number of input tokens needed, which is vital in latency-sensitive environments.
>
> We are committed to the ongoing enhancement of LoraHub and believe it has considerable potential across various applications. The insights from these initial challenges are invaluable, aiding in the enhancement of LoraHub's applicability and performance across a broader task spectrum. We value this opportunity to further refine and optimize LoraHub, ensuring it reaches its full potential in practical settings.
>
> > The number of LoRA modules
>
> We adopted this strategy primarily for computational efficiency. Increasing the number of LoRA modules enhances the dimensionality of the optimization problem, significantly affecting the time needed to find an optimal configuration. To manage complexity and balance performance with computational feasibility, we chose to use a subset of 20 modules.
>
> This method is supported by our findings in Appendix H, where we investigated the impact of increasing the number of optimization steps relative to the number of modules. These experiments indicated that although it is feasible to scale up the number of modules, this requires more optimization time, emphasizing the computational trade-offs involved.
>
> Using a fixed number of modules also demonstrates the stability and robustness of our approach under controlled conditions, essential for evaluating the effectiveness of the LoraHub method in practical applications. This strategy allows us to systematically assess the impact of module selection and composition without the overwhelming computational overhead associated with a larger set of variables.

---

### Official Review · Reviewer_7Jht · 2024-05-11

**Rating:** 6
**Confidence:** 4
**Ethics Flag:** 1

**Summary:**

This paper investigates the issue of efficiently combining multiple LoRAs for effective cross-task generalization. The proposed method, LoraHub, dynamically integrates multiple LoRA modules—each trained on distinct tasks—to enhance performance on new, unseen tasks. LoraHub is designed to efficiently integrate these modules without requiring human intervention, additional parameters, or gradients. Further experiments conducted using the BBH benchmark demonstrate that LoraHub can emulate in-context learning in few-shot scenarios, effectively eliminating the need for in-context examples during inference. Additionally, this approach encourages community-driven model sharing, which could significantly boost progress in general intelligence applications and the deployment of large language models in production environments.

**Questions To Authors:**

See reasons to reject.

**Reasons To Accept:**

1.	This paper introduces a novel method for achieving effective cross-task generalization with few-shot samples by reusing and combining existing LoRA modules for black-box optimization.
2.	Experimental results on the BBH benchmark indicate that LoraHub's performance rivals that of in-context learning, while processing fewer tokens and eliminating the need for gradients.
3.	The writing is clear and easy to follow.
4.	The authors have open-sourced a collection of diverse LoRAs, greatly benefiting further research within the community.

**Reasons To Reject:**

1.	The experimental results significantly lag behind other baselines, particularly when using many tasks' LoRAs with few-shot samples, where the model's performance remains weaker than that of in-context learning. Furthermore, the model substantially underperforms compared to other gradient-based methods. The authors need to consider more carefully the model's applicability and potential use cases.
2.	LoraHub randomly selects 20 LoRAs from the LoRA pool for combination, which seems arbitrary. This approach risks selecting LoRAs significantly mismatched to the downstream task while overlooking those from similar tasks, thus degrading model performance. Recent studies [1-2] have successfully employed a retriever mechanism to identify LoRAs that are more analogous to the downstream task for composition. Incorporating similar technology could refine LoraHub's selection process, making it more effective.
3.	The authors point out that gradient-based methods are resource-intensive. Therefore, it is crucial to quantify the computational resource consumption of different methods in the experiments to highlight LoraHub's advantages. This comparison was neglected in the conducted experiments.

[1] Chronopoulou, Alexandra, et al. "Adaptersoup: Weight averaging to improve generalization of pretrained language models." arXiv preprint arXiv:2302.07027 (2023).

[2] Zhao, Ziyu, et al. "LoraRetriever: Input-Aware LoRA Retrieval and Composition for Mixed Tasks in the Wild." arXiv preprint arXiv:2402.09997 (2024).

---

> ### Author Rebuttal · Authors · 2024-05-30
>
> > Model's applicability and potential use cases
>
> Thank you for pointing out the performance differences between LoraHub and other methods like ICL and gradient-based methods. We acknowledge the concerns regarding observed performance gaps in our experiments.
>
> LoraHub offers distinct advantages with its gray-box approach that requires only model output logits, avoiding the need for access to model weights and extensive GPU memory. This allows LoraHub to function efficiently in inference-only mode, ideal for CPU-only environments, simplifying deployment and reducing computational demands. Furthermore, as documented in Appendix B, LoraHub achieves a peak performance of 41.2 on the BBH benchmark, exceeding ICL's 38.4 and all other baselines except full fine-tuning, underscoring its potential superior performance.
>
> Your feedback is invaluable in our ongoing efforts to refine LoraHub, enhancing its performance and extending its practical applications to meet real-world needs.
>
> > Retriever mechanism
>
> We appreciate your critical insights regarding our LoRA module selection method in LoraHub. We understand your concerns about the apparent arbitrariness of our current selection process and its potential impact on downstream task performance.
>
> In response, we have introduced $\rm LoraHub_{filter}$ in Appendix E. This new approach enhances the selection process by identifying the top 20 LoRA module candidates that exhibit the lowest loss on a set of few-shot examples, helping to standardize and improve selection accuracy. Our tests indicate an improvement in average performance from 34.7 to 35.4, demonstrating the benefits of a more sophisticated retriever mechanism for module selection.
>
> > Comparison of resource requirement
>
> Thank you for emphasizing the importance of quantifying computational resource consumption in our studies. Our research primarily targets memory usage, as time requirements can vary with different epochs, dataset sizes, and GPU specifications.
>
> We follow to the memory test settings from the LoRA-FA study for an accurate benchmark. In this context, full fine-tuning required about **40GB** of memory, whereas LoRA fine-tuning used around **34GB**. Remarkably, LoraHub only utilized about **5GB** of memory, illustrating its efficiency due to the inference-only mode, which eliminates the need for storing gradients and optimization states. This comparison will be included in the revised version to clearly highlight LoraHub's resource efficiency.

---

### Official Review · Reviewer_dLQP · 2024-05-13

**Rating:** 5
**Confidence:** 4
**Ethics Flag:** 1

**Summary:**

The paper operates in the task generalization setting, where information (parameters) from seen tasks are used to efficiently generalize to downstream, unseen tasks. This task generalization setting have long been explored in the past but remains (imho) a relevant and interesting problem. Many related papers exist in the literature: the most related papers in this setting are (amongst others) SpoT and Polytropon. The novelty of this paper with respect to these papers is rather small: the authors use LoRAs instead of soft-prompts (as in SPoT) (minor), but use gradient free optimization for task adaptation (major novelty?). It is not clear to me however what is exactly the advantage of gradient-free optimization, i.e. the combination weights and adapters parameters themselves might be very cheaply optimized via gradient-descent on the downstream examples. The paper makes some nice ablations in the appendix and goes a little bit towards selecting best modules to combine or filtering modules. I think some more work in this direction might give a bit more novelty to this paper. I think in its current status this is quite a borderline paper.

**Questions To Authors:**

You might fix the few typos in the appendix:

NON-INSTRCUTION-TUNED -> Non-Instruction Tuned
Comparsion -> Comparison

**Reasons To Accept:**

- The paper is clear and very easy to read
- Operates in a very well-defined setting and therefore makes a focused contribution
- Ablations are nice
- This paper has already had quite some impact in democratizing the "parameter reuse" paradigm in the community (been in arxiv since a while)

**Reasons To Reject:**

- Novelty is somewhat limited with respect to existing works (MHR, SPoT, Polytropon)
- Performance on BBH is quite low (uses T5 model), so it might be difficult to get insights from ablations
- The justification for using gradient-free optimization appears a bit weak, I would love to see comparisons to gradient-based optimization of the LoRA weights (performance vs time)

---

> ### Author Rebuttal · Authors · 2024-05-30
>
> > Novelty is somewhat limited with respect to existing works (MHR, SPoT, Polytropon)
>
> Thank you for highlighting these related works. We appreciate your insights and will ensure to include these references in the revised version. LoraHub employs a unique gray-box method that uses only model output logits, unlike the gradient methods applicable mainly to open-source models in previous work. Furthermore, LoraHub functions solely during inference without requiring fine-tuning, facilitating efficient learning on CPU-only machines.
>
> > Performance on BBH is quite low (uses T5 model), so it might be difficult to get insights from ablations
>
> Thank you for your comments on the BBH metric performance. We recognize the concerns about the challenges in deriving insights from ablation studies with lower performance on the T5 model. However, our choice of the FLAN-T5 models as the foundation for our experiments is deliberate and strategic.
>
> The FLAN-T5 series is highly popular on HuggingFace, known for its proficiency in zero-shot and few-shot learning scenarios and robust problem-solving capabilities. These traits make it an excellent baseline for demonstrating the effectiveness of our methods, particularly in terms of learning efficiency and adaptability.
>
> Using FLAN-T5 also ensures a fair comparison in zero-shot scenarios. Our integration of LoRA modules, trained solely on the FLAN collection, allows us to maintain consistent conditions across experiments. This approach prevents any potential bias that might arise from introducing additional training data, which could unfairly benefit our method.
>
> Moreover, we have included results from FLAN-T5-XL, a 3B model, in the Appendix to further validate the generalizability of LoraHub across model scales.
>
> > The justification for using gradient-free optimization appears a bit weak, I would love to see comparisons to gradient-based optimization of the LoRA weights
>
> Thank you for your comment on the justification for using gradient-free optimization. We understand your interest in seeing a direct comparison between gradient-free and gradient-based optimization of the LoRA weights. Due to the constraints of our timeline, we were unable to include this specific comparison in the current version of our research. However, we acknowledge the value such a comparison would add to our findings and will incorporate the analysis into the revised version.
>
> Thank you for reminding and we will fix these typos in the revised version.

---

### Decision · Program_Chairs · 2024-07-10

**Decision:**

Accept

**Comment:**

The paper proposes a simple and intuitive framework for combining LoRA models by applying zero-order optimization on weights of these LORA additions to minimize loss.
The problem is important, but the solution is not particularly novel or surprising, yet the reviewers liked the paper. I therefore recommend acceptance pending space.